# Energy Recovery of Multiple Charge Sharing Events in Room Temperature Semiconductor Pixel Detectors

**DOI:** 10.3390/s21113669

**Published:** 2021-05-25

**Authors:** Antonino Buttacavoli, Gaetano Gerardi, Fabio Principato, Marcello Mirabello, Donato Cascio, Giuseppe Raso, Manuele Bettelli, Andrea Zappettini, Paul Seller, Matthew C. Veale, Leonardo Abbene

**Affiliations:** 1Department of Physics and Chemistry (DiFC)-Emilio Segrè, University of Palermo, Viale delle Scienze, Edificio 18, 90128 Palermo, Italy; antonino.buttacavoli@unipa.it (A.B.); gaetano.gerardi@unipa.it (G.G.); fabio.principato@unipa.it (F.P.); marcello.mirabello@unipa.it (M.M.); donato.cascio@unipa.it (D.C.); giuseppe.raso@unipa.it (G.R.); 2IMEM/CNR, Parco Area delle Scienze 37/A, 43100 Parma, Italy; manuele.bettelli@imem.cnr.it (M.B.); andrea.zappettini@imem.cnr.it (A.Z.); 3Science and Technology Facilities Council, Rutherford Appleton Laboratory, Chilton OX11 0QX, UK; paul.seller@stfc.ac.uk (P.S.); matthew.veale@stfc.ac.uk (M.C.V.)

**Keywords:** CZT pixel detectors, charge sharing, charge-sharing correction, semiconductor pixel detectors

## Abstract

Multiple coincidence events from charge-sharing and fluorescent cross-talk are typical drawbacks in room-temperature semiconductor pixel detectors. The mitigation of these distortions in the measured energy spectra, using charge-sharing discrimination (CSD) and charge-sharing addition (CSA) techniques, is always a trade-off between counting efficiency and energy resolution. The energy recovery of multiple coincidence events is still challenging due to the presence of charge losses after CSA. In this work, we will present original techniques able to correct charge losses after CSA even when multiple pixels are involved. Sub-millimeter cadmium–zinc–telluride (CdZnTe or CZT) pixel detectors were investigated with both uncollimated radiation sources and collimated synchrotron X rays, at energies below and above the K-shell absorption energy of the CZT material. These activities are in the framework of an international collaboration on the development of energy-resolved photon counting (ERPC) systems for spectroscopic X-ray imaging up to 150 keV.

## 1. Introduction

A large variety of room temperature semiconductor detectors (RTSDs) with sub-millimeter pixel electrodes was widely proposed for the next generation X-ray and gamma ray spectroscopic imagers [1,2,3,4,5,6]. Generally, RTSDs are represented by X-ray and gamma ray detectors based on high-Z and wide-bandgap compound semiconductors, with the goal to measure high-resolution energy spectra near room-temperature conditions. Currently, thin cadmium telluride (CdTe) and cadmium–zinc–telluride (CdZnTe or CZT) pixel detectors, with thickness up to 3 mm are considered the best choice up to 150 keV [7,8], while thicker detectors (up to 15 mm), based on CZT [9,10,11,12] and thallium bromide (TlBr) [13], are very appealing up to 1 MeV. RTSDs with small pixels represent a key choice for energy resolution improvements, which is in agreement with the *small pixel effect* [7] and potentially to obtain high spatial resolution. However, the presence of multiple coincidence events among neighboring pixels, due to charge-sharing and cross-talk phenomena [14,15,16], often results in spectroscopic and spatial degradations. A single interacting photon can induce charges in two or more pixels, generating multiple coincidence events with multiplicity *m* ≥ 2. Typically, the energy of these events is recovered by using charge-sharing addition (CSA) techniques, which consist of summing the energies of the coincidence events. However, as widely shown in both CdTe and CZT pixel detectors, charge losses at the inter-pixel gap often create incomplete energy recovery after CSA [17,18,19]. Recently, an energy-recovery technique for double charge shared events (*m* = 2) was proposed by our group [20] and successfully applied to several CZT and CdTe pixel detectors [20,21,22,23,24,25]. Despite the benefits on detector performance, this technique can only be applied to shared events involving only two adjacent pixels (*m* = 2) and, therefore, multiple coincidences with *m* > 2 and coincidences with diagonal pixels are rejected from the measured energy spectra. In order to recover the energy of multiple coincidence events and enhance the counting efficiency, we developed new correction techniques able to use all coincidence events in the measured energy spectra.

In this work, we will present new energy-recovery techniques for charge loss compensation after the application of CSA in CZT pixel detectors. Experimental investigations, taking into account the multiplicity of the coincidence events, were performed with both uncollimated and collimated synchrotron X-ray sources.

## 2. Charge-Sharing, Cross-Talk Phenomena and Correction Techniques

Intense investigations have been made on the effects of charge-sharing and cross-talk phenomena in CZT and CdTe pixel detectors [14,15,16,17,18,19,20,21,22,23,24,25]. Charge-sharing events are generated by the splitting of the charge cloud created by a single interacting event and collected by two or more pixels. The charge cloud collected by pixels is mainly due to the drifting electrons, which is in agreement with the *small pixel effect* [7]. The spatial region covered by the broadened electron cloud depends upon charge diffusion, Coulomb charge repulsion, K-shell X-ray fluorescence, and Compton scattering. Figure 1 shows the simulated evolution of the electron charge cloud size (FWHM) in a CZT detector vs. the drift distance, which is generated by photoelectrons in the energy range of 25–662 keV. The simulation involves the physical processes of Coulomb repulsion and charge diffusion [26]. It consists of three main procedures: (i) the radiation–semiconductor interaction with Monte Carlo methods (Geant4), (ii) electric and weighting field calculation by the finite element method (FEM) with COMSOL Multiphysics, and (iii) the calculation of the charge carrier transport and pulse formation in a MATLAB environment. The details of the simulation procedures are reported in a previous work [26]. The results are presented at two different electric field configurations: (a) 5000 V/cm representing the current state-of-art electric field in CZT detectors, while (b) the electric field of 10,000 V/cm was recently obtained in new high-bias voltage CZT pixel detectors [23,27]. For example, at 60 keV, the size of the charge cloud is about 60 μm and 45 μm over a drift distance of 2 mm, with 5000 V/cm and 10,000 V/cm, respectively.

At energies greater than the K-shell absorption energy of the CZT material (26.7, 9.7, and 31.8 keV for Cd, Zn, and Te, respectively), the broadening of the charge cloud size can be also increased by the presence of fluorescent X rays, whose emission probability is very high in CZT materials (≈85% of all photoelectric absorptions) [25,28]. The emission of fluorescent X rays of 23.2 keV (Cd-K_α1_) and 27.5 keV (Te-K_α1_) is more probable, with attenuation lengths of 116 and 69 μm, respectively. As shown in our previous works [19,20], 2 mm thick detectors with pixel pitches of 250 μm and an inter-pixel gap of 50 μm showed an increase of the coincidence percentage from 50% to 80% at 22 keV and 60 keV, respectively; this demonstrates as fluorescent X rays, present at 60 keV, give a great contribution to charge sharing.

Moreover, fluorescent X rays, escaping from the pixels, can also produce cross-talk events between pixels. Cross-talk events can be given by the *collected-charge pulses*; i.e., they are created by the charge carriers really collected by the pixels (e.g., fluorescent X rays) or by *induced-charge pulses* generated on neighboring non-collecting pixels [9,16,18]. These pulses, also called *transient pulses*, are due to the weighting potential cross-talk [9,10,11,18,20]; they are characterized by both positive and negative polarities and different shapes [9]. In our investigated energy range (up to 140 keV), a very low number of these transient pulses was detected. This is due to the low investigated photon energies that produce very small transient pulses, which are often below our detection energy threshold (4 keV). Transient pulses are clearly visible at higher energies (e.g., at 662 keV) [9,10]; these pulses measured in temporal coincidence with the collected pulses are strongly used for both spatial and energy resolution improvements in gamma-ray detectors [9,10].

The effects of charge-sharing and fluorescence cross-talk in the measured energy spectra are typically represented by a worsening of the energy resolution, the presence of low energy tailing below the photopeaks, fluorescence and associated escape peaks, and an increase of the low-energy background. The state-of-art on the mitigation of these effects is represented by the simple rejection of charge-sharing events (CSD: charge-sharing discrimination), which is detected in temporal coincidence with the events of the neighboring pixels; however, the high number of rejected events, often greater than 50% of all detected events, gives a strong reduction of the throughput and the counting efficiency of the detectors. To avoid this, the rejected events after CSD can be recovered through the charge-sharing addition (CSA) technique, which consists of summing the energies of the coincidence events [16,17,18]. Unfortunately, as documented in the literature [17,18,19,20,21], the energy recovered after CSA is often lower than true photon energy, which is due to the presence of charge losses near the region of inter-pixel gap. These losses are related to the presence of distorted electric field lines at the inter-pixel gap [14]. In fact, due to the surface conductivity at the inter-pixel gap, the electric field lines can intersect the surface of the inter-pixel gaps, where some charges can be trapped [14,20,23,24,25]. Generally, the charge losses depend on the interaction position within the gap, with high effects near the center.

Recently, an interesting technique for charge loss recovery after CSA was proposed by our group [20]. This technique, involving double-shared events (*m = 2*) between adjacent pixels, is based on the modeling of the charge losses vs. the interaction position within the inter-pixel gap. In particular, we modeled the relation between the summed energy *E_CSA_* for two adjacent pixels (*m* = 2), which are affected by the energy deficit, and the charge-sharing ratio *R*. As is well known, the charge-sharing ratio *R*, calculated from the ratio between the energies of two adjacent pixels *R* = (*pixel A* − *pixel B*)/(*pixel A* + *pixel B*), follows the interaction position of the events in the inter-pixel gap. Two key features characterize this technique: (i) first, the modeling does not depend on the photon energy, but it is only related to the physical and geometrical characteristics of the electrode layout; (ii) second, it can be easily obtained even with uncollimated photon beams. This approach was successfully applied to several CZT and CdTe pixel detectors [20,21,22,23,24,25] with improvements in energy resolution and counting efficiency. However, since this technique is applied to shared events involving only two adjacent pixels (*m* = 2), multiple coincidences with *m* > 2 are still rejected from the measured energy spectra.

The goal of this work is to propose new approaches for the correction of the energy of multiple coincidence events.

## 3. Detectors and Electronics

Charge-sharing investigations were performed on several CZT pixel detectors. We used CZT detectors characterized by the same anode layout but with different CZT crystals and thicknesses. The anode layout is characterized by four arrays of 3 × 3 pixels with pixel pitches of 500 and 250 µm (inter-pixel gap of 50 µm), while the cathode is a planar electrode (Figure 2). Different CZT crystals were used. Some detectors are based on the traveling heater method (THM) [19,20,21] and boron oxide encapsulated vertical Bridgman (B-VB) [23,27,29] CZT crystals. In addition to the standard or low flux LF-THM CZT crystals, we also tested high-flux HF-THM CZT crystals [20,30], which were recently produced by Redlen Technologies and characterized by enhanced hole charge transport properties to minimize radiation-induced polarization at high fluxes. The detectors were flip-chip bonded directly to analog charge-sensitive preamplifiers (CSPs) and processed by using digital pulse processing electronics [17,19]. The analog CSPs were based on a fast- and low-noise application specific integrated circuit (PIXIE ASIC) developed at RAL (Didcot, UK) [31]. The output waveforms from the CSPs are digitized and processed on-line by a 16-channel digital electronics, which were developed at DiFC of University of Palermo (Italy) [17,19]. The digital electronics is based on commercial digitizers (DT5724, 16-bit, 100 MS/s, CAEN SpA, Italy; http://www.caen.it, accessed on 24 May 2021), where an original firmware was uploaded [32]. Uncollimated radiation sources (^109^Cd, ^241^Am, ^57^Co) and collimated synchrotron X-rays (25 and 40 keV) were used for the measurements. In particular, collimated micro-beams were also used at the B16 test beamline at the Diamond Light Source synchrotron (Didcot, U.K.; http://www.diamond.ac.uk/Beamlines/Materials/B16, accessed on 24 May 2021). All detectors showed the effects of charge losses after CSA, which were successfully recovered after the application of the new techniques presented in the next sections. To simplify the presentation, we preferred to show the results of charge-sharing investigations for only one detector (2 mm thick HF-THM CZT detector) and some correction results even for the others. Generally, the detectors are characterized by room-temperature energy resolution less than 2 keV at 60 keV.

## 4. Multiple Coincidence Measurements and Corrections

In this section, we will present the results from time coincidence measurements and the effects of charge-sharing and fluorescence cross-talk in the energy spectra. Special attention will be given to the multiplicity *m* of coincidence events and the related energy recovery techniques. The coincidence measurements are always referred to the coincidence events of the central pixel with the eight neighboring pixels. We will start with the presentation of the energy spectra typically showed after charge-sharing investigations. In particular, Figure 3 presents three different energy spectra of uncollimated 59.5 keV photons from ^241^Am source, which are related to the central pixel of the 250 μm array of the 2 mm thick CZT detector. The black line represents the energy spectrum of all events (raw spectrum), and the red line is the spectrum of the events in temporal coincidence with all eight neighboring pixels.

The percentage of coincidence events is very high (79%), and the typical distortions related to charge-sharing and fluorescent cross-talk are clearly visible: the fluorescent peaks at 23.2 and 27.5 keV, the escape peaks at 36.3 and 32 keV, the low-energy background and tailing. The blue line is the spectrum after charge-sharing discrimination (CSD), i.e., after the rejection of coincidence events (79%). The coincidence analysis was performed with a coincidence time window (CTW) of 200 ns, ensuring the detection of all events; typically, about 90% of coincidence events are detected in a CTW of 30 ns. The CSD works well, allowing the rejection of all charge-sharing distortions, even if it produces a strong reduction of the pixel throughput and counting efficiency (79% rejected events). The tailing below the main photopeak is not mitigated; these events are not in temporal coincidence with neighboring pixels due to their energies being below the detection energy threshold (4 keV). Moreover, escape peaks are also present in the spectra even after CSD due to fluorescence events escaping from the cathode or absorbed on the guard-ring.

The multiplicity *m* of the coincidence events, i.e., the number of the involved pixels within a coincidence detection, was also measured (Figure 4). We estimated the multiplicity of the events of the central pixel (pixel no. 5) for both arrays and at different energies: ^109^Cd (main energy line at 22.1 keV), ^241^Am (main energy line at 59.5 keV), and ^57^Co (main energy line at 122.1 keV). Generally, the presence of coincidence events with *m >* 2 is increased for energies (^241^Am and ^57^Co sources) greater than the K-shell absorption energy of the CZT material, due to the fluorescence X-ray events.

The analysis of multiplicity was also performed on a sub-pixel level with collimated synchrotron X-ray beams. We measured the multiplicity of the coincidence events between two adjacent pixels at different beam positions. In particular, microscale line scans between the centers of the adjacent pixels were performed with collimated (10 μm × 10 μm) synchrotron X-ray beams at energies below (25 keV) and above (40 keV) the K-shell absorption energy of the CZT material. During the line scanning between the two pixels, we acquired, at each beam position (position steps of 25 μm), the data from all nine pixels of the investigated array. Figure 5 shows an overview of the multiplicity *m* vs. the beam position for two different line scans. The line scanning near the central region of both pixels (Figure 5a,c) highlights the presence of coincidence events mainly near the inter-pixel gap, with a spatial extension beyond the gap (50 μm) at 40 keV, due to fluorescence events; double coincidence events (*m* = 2) represent the dominant contribution to the overall coincidence events. Near the edge region of the pixels (Figure 5b,d), double coincidence events are present over all the pixel area; the number of multiple coincidence events (*m* > 2) is strongly increased near the inter-pixel gap, which is due to both charge-sharing and fluorescence cross-talk. In the following, we will investigate the features of the various multiplicities, presenting dedicated correction techniques.

### 4.1. Coincidence Events with Multiplicity m = 2

The predominant contribution to the overall coincidence events is represented by the events with multiplicity *m* = 2, as clearly shown in Figure 4. Double coincidence events are mainly due to charge-shared events near the inter-pixel gap, fluorescent cross-talk, and mixed shared/fluorescent events. Pictures of typical fluorescent cross-talk and charge shared events are shown in Figure 6a,b. To better highlight the different effects of the two main contributes, we analyzed the double coincidences of two adjacent pixels for a collimated irradiation (10 × 10 μm^2^) with synchrotron X rays at the center of one of the two pixels (Figure 6). Here, the energy spectra of the central pixel (Figure 6c) and the adjacent pixel (Figure 6d) are shown. All events of the adjacent pixel (green pixel no. 4) are in double temporal coincidence with events of the central pixel (black pixel no. 5); the shared events and fluorescent events are clearly visible, and the results after charge-sharing addition (CSA) are presented in Figure 7. The effects of charge-sharing and fluorescent cross-talk events after CSA are well distinguished. The energy of fluorescent cross events is fully recovered after CSA, as shown by the energy peak and the two kinks at *R* = 0.160 and *R* = 0.375 related to the 23.2 and 27.5 keV fluorescent X-rays, respectively; while charge losses characterized the double-shared events after CSA, which are highlighted by a reduction of the centroid of the main peak.

To recover the energy of double coincidences, we proposed two separate procedures for adjacent and diagonal pixels. The double coincidence events between adjacent pixels are mainly dominated by charge-shared events and by a small contribute of fluorescent cross-talk events. The energy of double shared events can be recovered through the correction technique, termed *double charge-sharing correction* (*double CSC*), as presented in our previous work [20].

It consists of the modeling of the 2D distribution of the energy after CSA (*E_CSA_*) vs. the charge-sharing ratio *R*, as shown in Figure 8a for uncollimated ^109^Cd source. This technique allows the energy reconstruction of double shared events from uncollimated and poly-energetic sources [20]. Figure 8b presents the results after the application of standard CSA and double CSC. After double CSC, the energy loss is recovered, and the energy resolution is also improved. At energies greater than the K-shell absorption energy of the CZT material, the double coincidence events between adjacent pixels also contain fluorescent cross-talk events, which can be easily corrected after standard CSA (Figure 7). The selection of these events (fluorescent event and escape peak event) is simple for mono-energetic X-ray sources, but it is challenging for poly-energetic sources. The dedicated selection of these events, based on stripping procedures with simulated response functions [33], will be investigated further in the future.

Concerning the double coincidence events between diagonal pixels, we observed that they are due to pure fluorescence cross-talk events. This is clearly shown in Figure 9. In particular, the 2D scatter plot of Figure 9a highlights the energy recovery of the energy after standard CSA at about *R* = ±0.22 and *R* = ±0.076, which are related to the fluorescent X rays of 23.2 and 27.5 keV, respectively (^241^Am source). The same result is also confirmed through the energy spectrum after CSA (Figure 9b). At energies below the K-shell absorption energy of CZT (e.g., by using^109^Cd source), no double coincidence events were observed between diagonal pixels.

### 4.2. Coincidence Events with Multiplicity m > 2

The results of Figure 4 point out the presence of coincidence events with multiplicity *m* > 2, especially for the 250 μm array. These events are created by mixed fluorescent/shared coincidence events at the inter-pixel gap. In particular, some triple coincidence events (i.e., *m* = 3) can be often obtained from a true quadruple coincidence, where the energy of the pulse of one pixel is below the detection energy threshold, e.g., of 4 keV in our case. The recovery of the energy of multiple coincidence events is still challenging. In Figure 10, we present the energy spectra of coincidence events after CSA at different multiplicities and energies. All energy spectra suffer from charge losses after CSA. However, the linear behavior of the summed energy *E_CSA_* with the true photon energy (Figure 10d) allows the recovery of the energy through a simple energy re-calibration procedure. We point out that the 4 keV intercept of the linear function for *m* = 3 (the red line of Figure 10d) highlights the possible measurement of triple coincidence events from true quadruple coincidences, where the energy of one of the four pixels is below the detection threshold (4 keV), and therefore, it is not detected.

Therefore, we applied the various proposed correction techniques for each related multiplicity. In particular, we applied, for the central pixel of 500/250 μm arrays, the *double CSC* technique on *m* = 2 events between adjacent pixels, the *standard CSA* for *m* = 2 events between diagonal pixels, and the *re-calibrated CSA* for both *m* = 3 and *m* = 4 events. The complete correction is termed *multiple CSC* technique. The results for two different CZT detectors are shown in Figure 11.

## 5. Conclusions

Original techniques that are able to correct the charge losses after charge-sharing addition (CSA) in CZT pixel detectors were presented. Different approaches were used for adjacent and diagonal pixels, taking into account the number of involved pixels (i.e., the multiplicity *m*). One approach, exploiting the relation between the summed energy after CSA and the charge-sharing ratio *R* of double coincidence events (*m* = 2), allowed the recovery of charge losses between adjacent pixels. A second technique, based on the linear behavior of charge losses after CSA with the true photon energy, was also implemented to reconstruct multiple coincidence events with *m* > 2. The energy of double coincidence events between diagonal pixels, mainly due to fluorescence cross-talk events, was successfully recovered after the standard CSA.

Results on different CZT pixel detectors showed improved counting efficiency compared to using only isolated events (i.e., after CSD) and improved energy resolution compared to the standard CSA techniques.

## Figures and Tables

**Figure 1 sensors-21-03669-f001:**
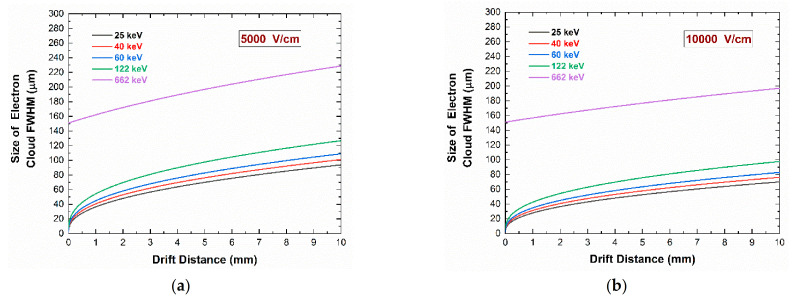
Evolution of electron cloud size (FWHM) vs. the drift distance in CZT detectors generated by photoelectrons in the energy range of 25–662 keV; Coulomb repulsion and charge diffusion are used in the simulation [26]. (**a**) Electric field of 5000 V/cm; (**b**) 10,000 V/cm.

**Figure 2 sensors-21-03669-f002:**
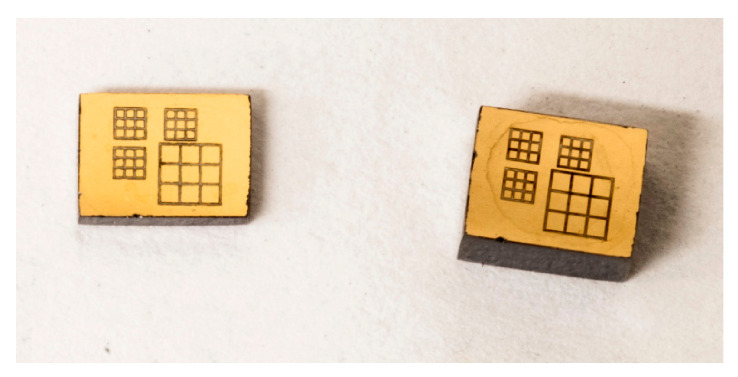
Anode layout of the investigated CZT pixel detectors. Each detector is characterized by four arrays of 3 × 3 pixels with pixel pitches of 500 and 250 μm.

**Figure 3 sensors-21-03669-f003:**
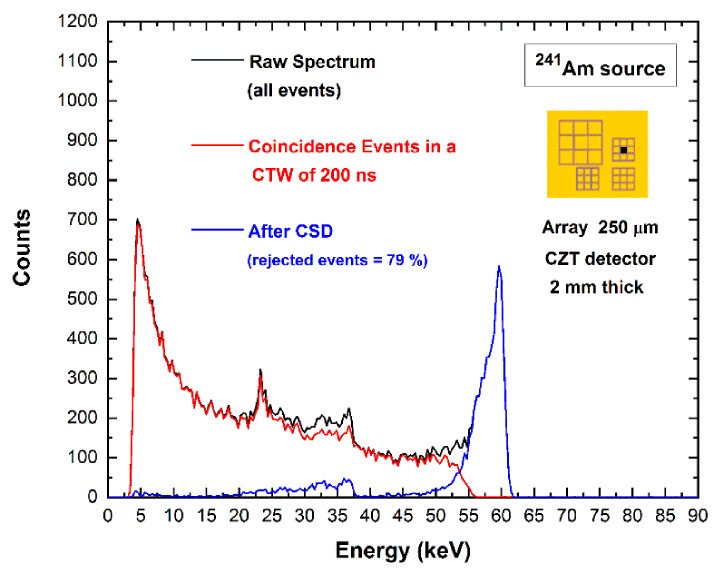
Charge-sharing measurements for the central pixel of the 250 μm array. The blue line represents the uncollimated ^241^Am spectrum after charge-sharing discrimination (CSD). The raw spectrum (black line) of all events and the spectrum of the coincidence events with all eight neighboring pixels (red line) are also shown. The yellow inset shows the layout of the anode of the detector.

**Figure 4 sensors-21-03669-f004:**
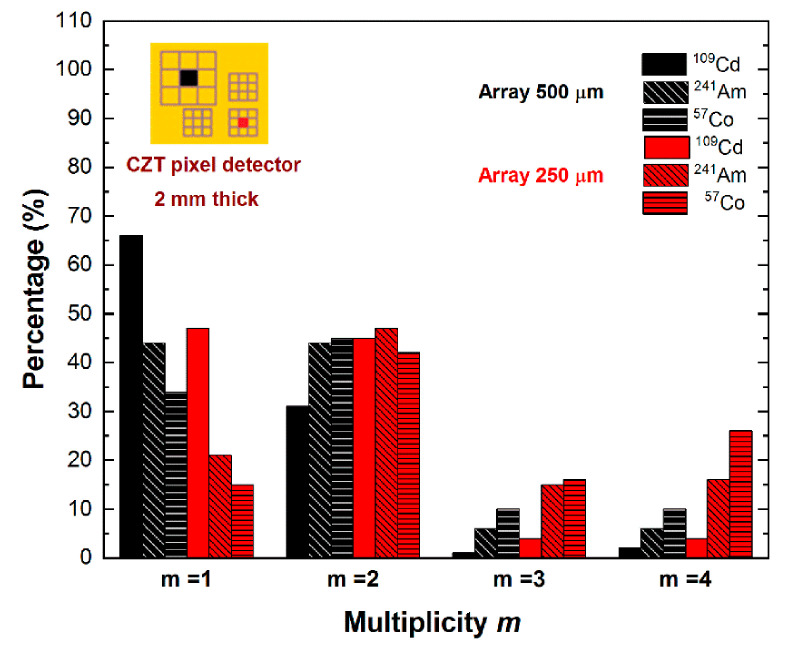
The multiplicity *m* of the coincidence events of the central pixels for the 250 and 500 μm arrays. The multiplicity *m* is referred to the number of pixels involved in a coincidence detection.

**Figure 5 sensors-21-03669-f005:**
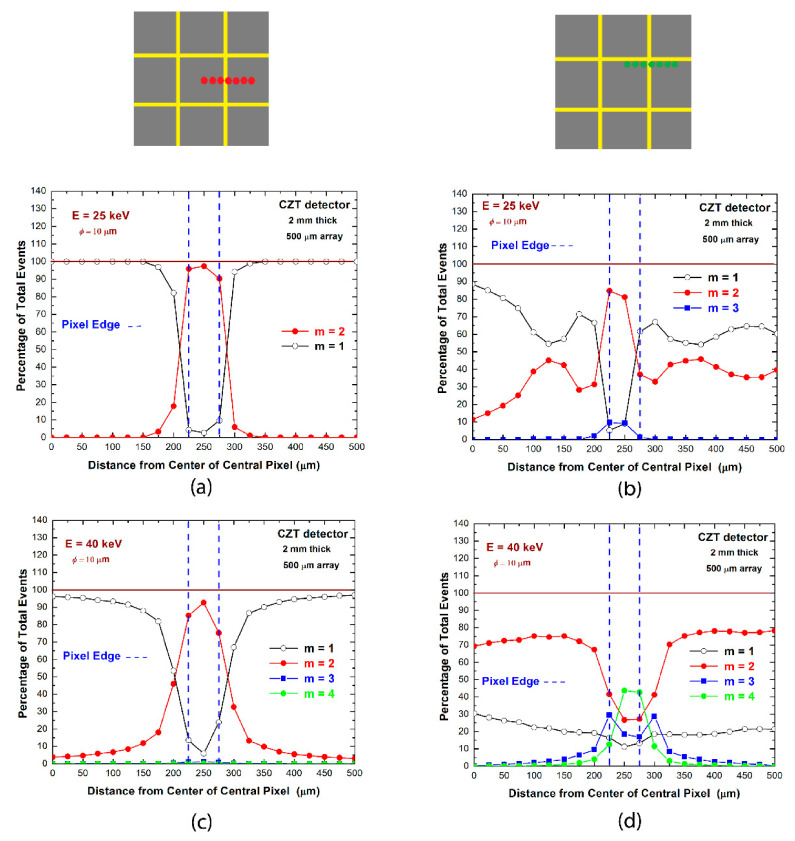
Results of microscale line scanning with synchrotron X-rays between the centers of two adjacent pixels; (**a**,**b**) at energy below (25 keV) and (**c**,**d**) above (40 keV) the K-shell absorption energy of the CZT material. The multiplicity *m* at various beam positions was measured: (**a**,**c**) near the central region of both pixels, (**b**,**d**) near the edge region.

**Figure 6 sensors-21-03669-f006:**
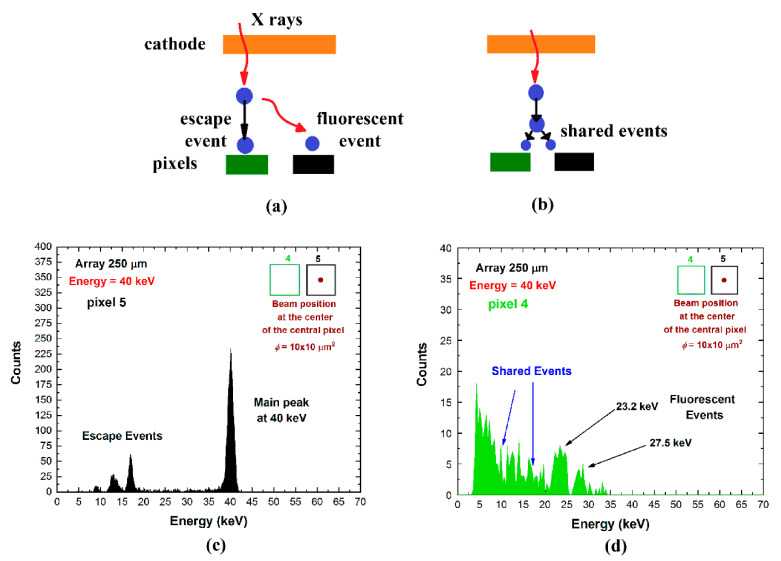
Measurements of fluorescence cross-talk (**a**) and charge-sharing (**b**) events. Energy spectra at 40 keV of two adjacent pixels: (**c**) pixel no. 5 and (**d**) pixel no. 4. We used a collimated X-ray synchrotron beam (10 × 10 μm^2^) interacting at the center of the central pixel (black pixel no.5). All events of the adjacent pixel (green pixel no. 4) are in temporal coincidence with the central pixel with multiplicity *m* = 2.

**Figure 7 sensors-21-03669-f007:**
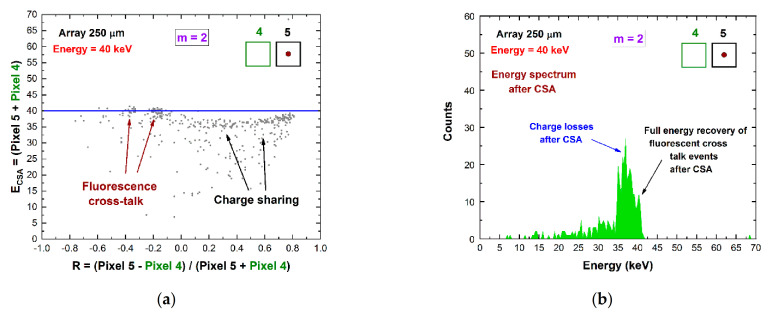
Experimental results after CSA for *m* = 2 coincidence events between adjacent pixels. (**a**) The 2D scatter plot of the energy *E_CSA_* after CSA versus the charge-sharing ratio *R*. (**b**) The energy spectrum after CSA.

**Figure 8 sensors-21-03669-f008:**
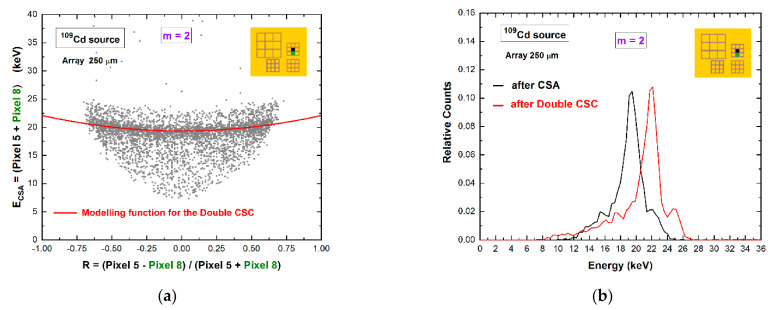
(**a**) Two-dimensional (2D) scatter plot of the summed energy of the coincidence events (*m* = 2) between pixel no. 5 and pixel no. 8, after CSA, versus the ratio *R* (uncollimated ^109^Cd source). The red line is the modeling function, a parabola with vertex at the point (0, *E_CSA_* (*0*)) [20], used to correct charge losses. (**b**) The energy spectra after CSA (black line) and after double CSC (red line). The complete recovery of the energy after double CSC and improvements of the energy resolution are clearly visible.

**Figure 9 sensors-21-03669-f009:**
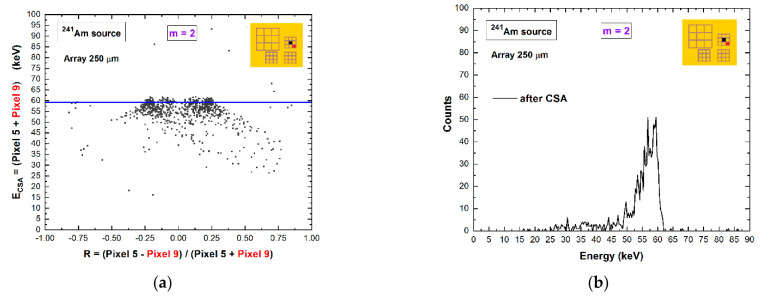
(**a**) Two-dimensional (2D) scatter plot of the summed energy of the coincidence events (*m* = 2) between diagonal pixels (no. 5 and pixel no. 9), after CSA, versus the ratio *R*. (**b**) The energy spectrum shows the absence of charge losses after CSA, demonstrating that the double coincidence events between diagonal pixels are mainly due to the fluorescent/escape events.

**Figure 10 sensors-21-03669-f010:**
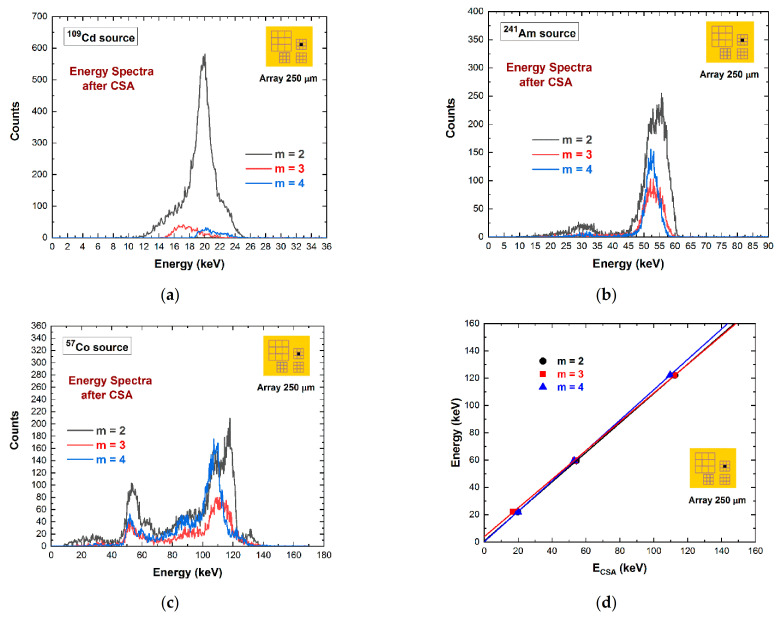
(**a**) ^109^Cd, (**b**) ^241^Am, and (**c**) ^57^Co spectra after CSA at various multiplicities. (**d**) The linearity of E_CSA_ with the true photon energy opens up to a correction of charge losses after CSA through a simple energy re-calibration.

**Figure 11 sensors-21-03669-f011:**
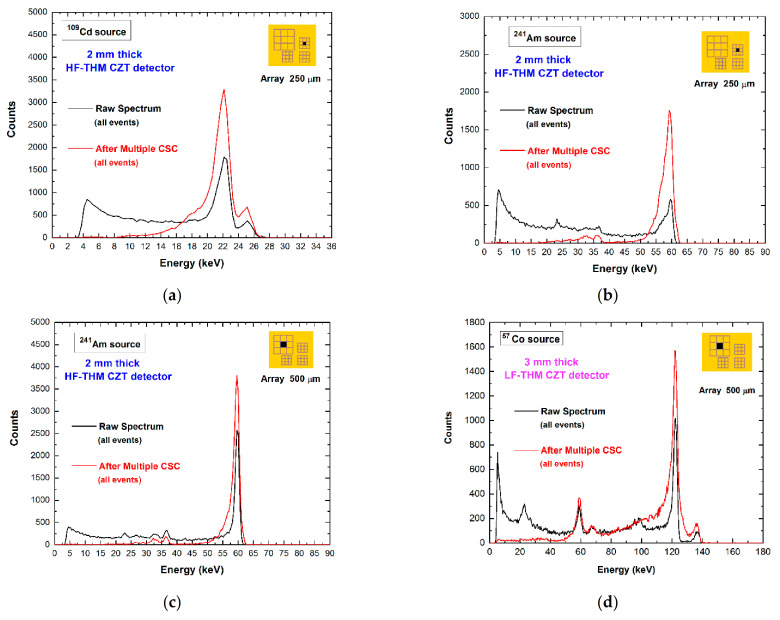
The raw energy spectra (black lines) and the corrected spectra (red lines) after the multiple charge-sharing correction (CSC) of the central pixel of 500/250 μm arrays for different detectors: (a–c) the high-flux HF-THM CZT detector, (d) the low-flux LF-THM CZT detector. We applied the double CSC technique on *m* = 2 events between adjacent pixels, the standard CSA for *m* = 2 events between diagonal pixels, and the re-calibrated CSA for both *m* = 3 and *m* = 4 events. The energy resolution was slightly improved.

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
