# Peer review of "Energy Recovery of Multiple Charge Sharing Events in Room Temperature Semiconductor Pixel Detectors"

_sensors, 2021, doi:10.3390/s21113669_

Round 1

Reviewer 1 Report

please see attached comments

Reviewer 2 Report

I would like to congratulate authors for a great work. The article is well prepared and it is easily to read. The main these is clear and the solution well described with adequate measurements. Readers can clearly make their own conclusions basing on the results. 

Pixels are a bit large though. The size around 100um or even lower would be a nice improvement to the study.

Reviewer 3 Report

The paper " Energy recovery of multiple charge sharing events in 2 room temperature semiconductor pixel detectors" describes a calibration method to combine the energy deposited in multiple pixels to determine the full X-ray energy.  The paper is for sure interesting to read and well written. The only pity is that the authors do not try to study in depth the signal formation to justify the findings, but try to find an algorithm that works with the data collected. 

  • Reproducibility of the results among the different sensor samples should be commented.
  • How well have you calibrated the pixels to start with?
  • As you mention coincidence in times all times, what is the time resolution of your detector? There is no doubt about the coincidence, they are real, but I am curious how large the time resolution is.
  • Maybe a cartoon explaining what happens when we have fluorescence photons, escaping photons etc could help the reader to follow.

Abstract: l.23 CZT never introduced to the reader before. It should be expressed as CdZTe the first time.

l.68: simulated, how?

l.89: the sentence does not seem correct in English and not too clear to the reader either. Do you want to say that as you go from 60keV from 22keV, you increase the coincidence counts, which must come from extra fluorescence?

Figure 7a: numbering over the pixels (4, 5) is needed 

Figure 8a: can you comment on the modelling function?

 Figure 10 c increase bin size 
